# Form-Stable Composite Phase Change Materials Based on Porous Copper–Graphene Heterostructures for Solar Thermal Energy Conversion and Storage

**DOI:** 10.3390/polym15244723

**Published:** 2023-12-16

**Authors:** Chao Chang, Bo Li, Baocai Fu, Xu Yang, Yulong Ji

**Affiliations:** Institute of Marine Engineering and Thermal Science, Marine Engineering College, Dalian Maritime University, Dalian 116026, China

**Keywords:** solar energy, phase change materials, graphene, thermal conductivity

## Abstract

Solar–thermal energy conversion and storage technology has attracted great interest in the past few decades. Phase change materials (PCMs), by storing and releasing solar energy, are able to effectively address the imbalance between energy supply and demand, but they still have the disadvantage of low thermal conductivity and leakage problems. In this work, new form-stable solar thermal storage materials by impregnating paraffin PCMs within porous copper–graphene (G–Cu) heterostructures were designed, which integrated high thermal conductivity, high solar energy absorption, and anti-leakage properties. In this new structure, graphene can directly absorb and store solar energy in the paraffin PCMs by means of phase change heat transfer. The porous structure provided good heat conduction, and the large surface area increased the loading capacity of solar thermal storage materials. The small pores and superhydrophobic surfaces of the modified porous G–Cu heterostructures effectively hindered the leakage issues during the phase change process. The experimental results exhibited that the thermal conductivity of the prepared form-stable PCM composites was up to 2.99 W/(m·K), and no leakage took place in the solar–thermal charging process. At last, we demonstrated that the PCM composites as an energy source were easily integrated with a thermoelectric chip to generate electric energy by absorbing and converting solar energy.

## 1. Introduction

To date, the continuous consumption of traditional fossil energy and global environmental pollution have caused a serious energy crisis. Developing different types of clean energy sources has become an essential approach to achieving sustainable development in human society [1]. Owing to its abundance, cleanliness, and low cost, solar energy as an alternative energy resource attracts intensive attention all over the world [2,3]. At present, solar energy utilization technology is mainly classified into solar thermal technologies and photovoltaic technologies. Compared with photovoltaic technologies, solar thermal technologies, which absorb solar radiation directly and convert it into heat to heat up liquid or air, have higher energy conversion efficiency, lower cost, and a longer service life [4,5,6]. In the current solar thermal systems, however, intermittency and instability deteriorate the efficiency of the systems. To address the mismatch between demand and supply of solar energy, solar thermal storage technologies, which absorb, store, and release vast quantities of thermal energy, are considered a promising method [7,8,9]. Solar thermal storage materials are the key to solar thermal storage systems, which include latent heat storage, sensible heat storage, and chemical reaction heat storage. Among them, the latent heat storage technologies that rely on the liquid–solid phase change material (PCM) have many advantages, such as large phase change enthalpy, high heat storage density, non-poisonous, and near constant solid–liquid transition temperature [10,11]. According to their chemical properties, the common PCMs are divided into two classes: inorganic PCMs—including molten salts, crystal salts, metals, and metal oxides—and organic PCMs, including paraffin, alcohols, fatty acids, and some polymers. In fact, conventional PCMs suffer from low thermal conductivity and leakage problems, which deteriorate their energy storage rate and seriously limit their practical applications [12,13,14].

To enhance the thermal conductivity of PCMs, many high thermal conductivity fillers have been added to PCMs, including expanded graphite [15,16,17], carbon nanotube [18,19], graphene [20,21,22], and metal foam [23,24,25,26]. The dispersion of high-conductive nanomaterials is regarded as one of the most efficient approaches to enhancing the thermal conductivity of base PCMs. Table 1 summarizes the typical composite PCMs for solar thermal storage. A large number of studies exhibited that dispersing these high-conductive additives in the PCM would effectively improve thermal conductivity and obtain excellent heat transfer performance. For example, Kumar et al. [27] investigated the effect of nano-Si_3_N_4_ particles on the thermal conductivity of the paraffin PCMs. The experimental results showed that the thermal conductivity of the composites was increased by 33.9% when the weight percentage of Si_3_N_4_ nanoparticles was 2%. Xie et al. [28] prepared a solar–thermal storage composite material by filling carbon fiber and graphite sheets and expanding graphite into organic PCMs. Compared to expanded graphite/organic PCMs, the thermal conductivity of this composite was up to 16.5 W/(m·K), which increased by ca. 24%. Shama et al. [29] synthesized promising solar thermal storage materials by filling commercial paraffin with CuO nanoparticles. It was found that the enhancement of thermal conductivity of the composite materials was over 17%, and the melting time was decreased by 22.22% as the concentration of CuO nanoparticles was 1%. Although these highly conductive nanoparticles had the potential to enhance the thermal conductivity of PCMs, an effective improvement of their thermal property of PCMs required further pressing the composite materials into a compact block [30,31]. In addition to high-conductive nanomaterials, embedding PCMs into porous materials was another common method to improve thermal conductivity by increasing the heat transfer area. For example, Zhang et al. [32] presented a PCM composite prepared by embedding PCMs into a three-dimensional diamond foam. The new PCM had a thermal conductivity of 6.70 W/(m·K), which was a great enhancement over pure PCMs.

On the other hand, many researchers have investigated various methods to restrict the leakage of PCMs. Metal foams with many advantages, such as a large specific surface, relatively high thermal conductivity, low bulk density, and high air permeability, have been widely used for enhancing the thermal performance of PCMs [33,34,35,36]. In the metal foams, the porous structure and skeleton connection enabled more PCMs to be embedded, which enhanced the thermal conductivity and heat storage capacity of PCMs. For example, Zhang et al. [37] reported shape-stabilized composite solar thermal storage materials, which were made of paraffin and PEG10000 as organic PCMs, copper foams as supporters, and carbon-based materials as surface modifiers. The thermal conductivity of this new shape-stabilized composite material was 1.04 W/(m·K), which indicated an increase of 300% compared with pure PCMs, and the solar–thermal conversion efficiency was 86.68%. Zheng et al. [38] prepared a novel form-stable solar thermal storage material that was prepared by impregnating paraffin PCMs into copper foams loaded with graphene aerogels. The thermal conductivity of this new composite material was about 3.0 W/(m·K), which was over 9.0 times higher than that of pure PCMs, and the solar–thermal energy conversion efficiency reached up to 97%. Zheng et al. [39] investigated the effect of copper foams on the melting behavior of paraffin through visual experiments. It was found that copper foam could reduce the melting time of paraffin by 20.5%, and when the heat position was at the top, the melting time was the longest compared with it heated on the left and at the bottom. Wang et al. [40] established a test platform to analyze the thermal properties of PCM composites, and the phase change process was investigated using numerical simulations. The results showed that porous copper foams with a porosity of 97.3% could effectively enhance the uniformity of PCM’s internal heat transfer while shortening its thermal storage time by over 40%. In addition to copper foams, El Idi et al. [41] adopted aluminum and nickel foams to prepare a paraffin–metal foam composite using a vacuum impregnation method. The experimental results exhibited that the thermal conductivity of the paraffin–aluminum foam composite was about 18.0 times higher than that of commercial paraffin, and the paraffin–nickel foam composite was about 6.0 times higher than that of commercial paraffin. Xiao et al. [42] reported a PCM composite which was prepared by embedding pure paraffin into nickel foams with various pore sizes and thermal porosities by the vacuum mothed. The thermal conductivity of the PCM composite prepared by nickel foams with a porosity of 90.61% reached up to 2.33 W/(m·K), which was nearly five times higher than that of pure paraffin. Therefore, the metal foams provided a promising way to improve the thermal conductivity of the paraffin and avoid the problem of leakage simultaneously.

**Table 1 polymers-15-04723-t001:** Properties of typical PCM composites.

Composites	Porosity	Thermal Conductivity (W/(m·K))
Expanded graphite–paraffin composite [15]	38.01%	2.45
Expanded graphite–hexadecane composite [17]	80%,	1.2402
Melamine–paraffin composite [21]	85.8%,	0.096
Copper–paraffin composite [37]	95%	1.04
Copper–paraffin composite [40]	97.3%	2.879
Nickel–paraffin composite [41]	95.2%	1.44
Nickel–paraffin composite [42]	90.61%	2.33
Copper–paraffin composite [43]	95.92%	1.439
Copper–paraffin composite [43]	97.59%	1.238
Nickel–myrtle alcohol composite [44]	97%	0.48

In this work, we presented a facile and direct method to prepare form-stable solar thermal storage materials via impregnating paraffin PCMs within porous copper–graphene (G–Cu) heterostructures, which integrated high thermal conductivity, high solar energy absorption, and anti-leakage properties. The porous G–Cu heterostructures were fabricated via sintering multilayer copper meshes at a high temperature, followed by graphene layers growing on it to obtain the ability of solar energy absorption, and then modified with a layer of polydimethylsiloxane (PDMS) to obtain a hydrophobic surface. The sintering porous structure covered with graphene layers provided efficient heat conduction, and the wide surface area increased the load capacity of solar thermal storage materials. The small pores and superhydrophobic surfaces of the modified porous G–Cu heterostructures effectively hindered the leakage issues during the phase change process of paraffin PCMs. Graphene layers were able to directly absorb solar radiation and convert it into thermal energy, which was stored in the paraffin PCMs by means of phase change heat transfer. In addition, we also verified that the novel composite materials based on G–Cu heterostructures were successfully used to power a thermoelectric chip and generate electric energy. Therefore, this work not only offered a new method to fabricate high-performance PCM composites but also integrated the processes of fast thermal response, solar–thermal conversion, and solar thermal energy storage.

## 2. Materials and Methods

### 2.1. Materials

Paraffin PCMs were brought from Shanghai Aladdin Reagent Co., Ltd. (Shanghai, China). Copper meshes (300-mesh) were purchased from Jiangsu Ju Cheng Mesh Co., Ltd. (Suzhou, China). Hydrochloric acid (HCl) was provided by Shanghai Lingfeng Chemical Reagent Co., Ltd. (Shanghai, China). Polydimethylsiloxane (PDMS) and a foaming agent (Sylgard 184) were ordered from Shenzhen SINWE Co., Ltd. (Shenzhen, China). High-purity methane and mixed gas (N_2_, 95%; H_2_, 5%) were brought from Newradar SPECIAL Gas Co., Ltd. (Dalian, China). The thermoelectric generator (SP1848-27145) was provided by Guangzhou ElecFans Electronic Co., Ltd. (Guangzhou, China).

### 2.2. Preparation of Copper–Graphene Heterostructures

The copper–graphene heterostructures were prepared by a high-temperature sintering method, which was further was used to synthesize the PCM composites, as shown in Figure 1. First, the copper mesh with 60-mesh was immersed in 4 mol/L HCl solution in order to remove its surface copper oxide, and then it was dried in an oven. The treated copper mesh was cut into small pieces to construct a cube with a length of 34 mm, a width of 20 mm, and a height of 20 mm. In this structure, there were five pieces of copper meshes as internal supporters and heat transfer paths. Then, the cube was loaded in a corundum boat and put into a tube furnace. The temperature gradually rose from room temperature to 900 °C with a temperature rise rate of 10 °C/min, and then it was held for 60 min under the N_2_-H_2_ mixed gas atmosphere. To obtain graphene nanoparticles, CH_4_ gas as the carbon source was introduced into the tube furnace and held for another 40 min. After the modification of graphene nanoparticles, the N_2_-H_2_ mixed gas replaced CH_4_ gas, and the reactor chamber was cooled down to room temperature naturally. The porous G–Cu heterostructures were further modified by a layer of PDMS to obtain hydrophobicity. Last, we adopted a vacuum impregnation method to prepare the PCM composites by immersing the modified G–Cu heterostructures in melted liquid paraffin PCMs.

### 2.3. Preparation of the Graphene–Copper–Paraffin Composites

To obtain surface hydrophobicity, the sintered copper–graphene heterostructures were first immersed in a 40 mL n-hexane solution which contained PDMS and curing agent with a ratio of 10:1. Then, the modified porous copper–graphene heterostructures were put in a drying oven with a temperature of 60 °C to cure the PDMS. Finally, the modified copper–graphene heterostructures were placed into the liquid melted paraffin for 5 min and cooled to room temperature to obtain the graphene–copper–paraffin composites. 

### 2.4. Measurement and Characterization

The morphology of the PCM composites was analyzed by field emission scanning electron microscopy (Czech TESCAN MIRA LMS, Brno, Czech Republic). The wettability of the copper–graphene heterostructures surface was measured by a contact angle measurement analyzer (Dataphysics OCA20, Stuttgart, Germany). The crystal quality of the composite was characterized by X-ray diffraction (XRD, Thermo Scientific ARL EQUINOX 3500, Waltham, MA, USA). The Raman spectrum was obtained by a Renishaw inVia Qontor confocal Raman microscope system (Gloucestershire, UK). The solar irradiation was produced by a solar simulator (CEL-PE300L-3A, Beijing, China), and a solar power meter (CEL-NP2000-2A, Beijing, China) was used to calibrate the solar power density. To measure the thermal conductivity, K-type thermocouples (Omega SMPWTT-K) were used to record the real-time temperature of the PCM composite, and the temperature signal was stored in a multichannel data acquisition system (Agilent 34970a, Agilent Technologies Inc., Santa Clara, CA, USA). The cooling block was connected to a cooling bath (Julabo Bilon Equipment, Seelbach, Germany), which provided circulating running cooling water with a consistent temperature of 10 °C. To monitor the solar charging process, an infrared camera (HM-TPH36-10VF/W HIKVISION, Hangzhou, China) was used to take time-sequential IR images.

## 3. Results and Discussion

The porous G–Cu heterostructures played a critical role in the solar energy storage process for the PCM composites. The graphene nanoparticles grown on the surface of the copper with excellent solar–thermal conversion capability were able to directly collect solar radiation and convert it into heat. Porous G–Cu heterostructures not only enabled the embedding of many PCMs but also efficiently captured the incident sunlight, thereby reducing scattering reflection. The G–Cu heterostructures with good skeleton connections offered an effective heat transfer path for solar thermal energy storage. Additionally, the capillary force generated by the hydrophobic surfaces and small size pores efficiently restricted the leakage of melted paraffin PCM during the phase change process. 

Figure 2a,b present the SEM images of the G–Cu heterostructures and untreated copper mesh at low and high magnifications, respectively. As shown, a layer of graphene nanoparticles is formed on the G–Cu heterostructures surface, resulting in a rough surface, as shown in Figure 2a. On the contrary, the untreated copper mesh surface presents relatively smooth, and the color is slightly darker than that of the G–Cu heterostructures surface, as shown in Figure 2b. To further analyze its structure, the X-ray diffraction (XRD) and the Raman spectrum are acquired from the outer surface under 532 nm laser excitation, as shown in Appendix A. From the XRD results in Appendix A, it can be seen that there are 2θ peaks at about 43°, 50°, and 74° which are well indexed to (111), (200), and (220) for Cu, respectively. The 2θ peak at 26.3° is indexed to the C (002), indicating that the composites only consist of graphene and copper, where graphene is composed of crystalline carbon atoms. From its Raman spectrum, we could see the typical graphene Raman peaks (G-band at ~1342 cm^−2^, D-band at ~1580 cm^−2^, and two-dimensional band at ~2680 cm^−2^) appearing on the surface, indicating the presence of graphene on its surface.

A water contact angle measurement was used to evaluate the wettability of the porous G–Cu heterostructures. To reduce the contact angle hysteresis influence caused by the water droplets, we placed the needle at a distance of 1 to 2 mm up to the test samples. A computer was used to control the machine to drop 2 μL of water on the surface of the samples. A calculator continuously shot with a high-power camera; meanwhile, a computer measured its contact angle automatically. During the test, the room temperature was 20 °C and the relative humidity was 30%. As shown in Figure 2c, the surface of the G–Cu heterostructure modified by PDMS presents hydrophobicity, and the measured contact angle is 132.3° after a water droplet is on the surface for 1 s, while the untreated copper mesh contact angle is only 72.4° (Figure 2d). Additionally, we calculated the porosity of the porous G–Cu heterostructures by comparing their weight and volume. The porosity of the G–Cu heterostructures reached up to 90.5%, meaning that it could embed more paraffin PCM. Both the rough surface and hydrophobic properties would promote embedding more paraffin PCM in the porous copper structure.

Figure 3a exhibits an optical photograph of the prepared solar thermal energy storage PCM composites. The PCM composites with a size of 34 mm × 20 mm × 20 mm present a dark black color. Form stability is another critical factor affecting the thermal performance of the solar thermal energy storage materials during the charging and discharging processes. In order to test the thermal stability, we put both the PCM composites and pure commercial paraffin on the same heating plate with a temperature of 80 °C. Figure 3a,b present the measurement results of the form stability of the PCM composites and pure paraffin PCM. As shown, after being heated for 10 min, the prepared PCM composites retain their original shape, and there is no melted liquid paraffin to be observed to flow out of the sample. In contrast, under the same heat condition, the commercial paraffin PCM block is completely melted into liquid within 60 s. The weight variation of our prepared G–Cu-based PCM composites is also measured before and after heating, as shown in Figure 3c. It is clearly seen that after 10 thermal cycles, no significant weight loss is observed, and the leakage of the prepared PCM composites is no more than 0.5%.

In addition to form stability, the thermal conductivity played a significant role in the thermal performance of the PCM composites. Based on Fourier’s law, we establish a simple measurement setup, and its measurement mechanism is shown in the inset in Figure 3d. The test system is made up of an electric heater, a constant temperature water bath, a data acquisition instrument, and two k-type thermocouples. We locate the PCM composites between the electric heater and the cooling block, and two thermocouples are used to monitor the temperature difference of the composites as the heating input increasing. To minimize heat loss, the whole test unit is covered with a layer of insulation foams. According to the measured temperatures, the thermal conductivity (*k*) of the PCM composites could be calculated from Equation (1) as follows:(1)$$Q_{in}=Ak\frac{∆T}{d}+Q_{ex}$$
where A, k, and d are the cross-sectional area, the thermal conductivity, and the thickness of the PCM composites, respectively. ∆T is the temperature difference between the top and bottom of the PCM composites. $Q_{ex}$ is the thermal loss from the electric heater to the atmosphere. Based on Equation (1), when $Q_{ex}$ is considered to be a constant value, the heating input $Q_{in}$ has a linear relationship with the temperature difference ∆T. The variation of heating input $Q_{in}$ of the PCM composites with temperature difference ∆T is shown in Figure 3d. Through the linear fitting of temperature differences at different heating inputs, the PCM composites’ thermal conductivity *k* is calculated to be 2.99 W/(m∙K).

In order to estimate solar–thermal conversion properties, we placed our prepared PCM composites and a pure PCM thermal pack under solar radiation. The solar radiation was generated by a solar simulator (CEL-PE300L-3A, Beijing, China), and a solar power meter (CEL-NP2000-2A, Beijing, China) was used to calibrate the solar power density. During the experiment, the solar power intensity was adjusted to 5 kW/m^2^ to achieve rapid solar charging. To reduce heat loss, the tested samples were placed on an insulating mat. We used an infrared camera to monitor the real-time temperature variation of the two samples. As shown in Figure 4a, the whole solar charging process of our developed PCM composites and pure paraffin samples lasts for 11 min. As shown, before the sunlight shining on the surface of the samples, the PCM composites and pure paraffin thermal pack are at the same temperature (20 °C). After solar charging for 6 min, it is clearly seen that the PCM composites are obviously heated to a high temperature, while the temperature of the pure paraffin thermal pack is still very low. Figure 4b presents the temperature variation of the pure paraffin thermal pack during the solar charging process. It should be noted that the temperature of the pure paraffin pack is always below 40 °C during the whole solar charging process, meaning that it is not melted even if it is charged for 11 min, which can be attributed to its low solar absorption and low thermal conductivity.

Figure 4c exhibits the temperature variation of the PCM composites during the solar charging process. Compared with the pure paraffin thermal pack, all the measured temperatures on various surfaces of our developed PCM composites almost coincided during the entire solar charging process. Based on the temperature variation curves, it is clearly seen that after solar charging for 4 min, the developed PCM composites start to enter a phase transition from solid to liquid phase. After the PCM composites are completely melted, the process of the solar charging is transformed into sensible heat storage instead of latent heat storage. As shown, compared with the pure paraffin thermal pack temperature, the PCM composites can be heated to over 60 °C. On the other hand, due to the low thermal conductivity, the maximum temperature difference between the top surface and bottom of the pure paraffin thermal pack is approximately 5.8 °C after solar charging for 11 min, as shown in Figure 4d. The unique solar–thermal conversion performance can be attributed to the high solar absorption of graphene nanoparticles grown on its surface. Additionally, our PCM composites have a relatively high thermal conductivity, which will facilitate the heat transfer during this solar charging process, thereby resulting in the sample being heated up uniformly. 

After confirming the high thermal performance, we further explored its application by combining the prepared PCM composites with a thermoelectric (TE) generator. The PCM composites can absorb solar energy directly and store the collected thermal energy in the PCMs, which will activate the TE chip to produce electricity energy by thermoelectric effect. Figure 5a exhibits the schematic of our designed solar–electric conversion system. In the TE generator, the hot side is connected to the PCM composites, and the cold side is connected to a cooler. To reduce the thermal resistance of the interfacial, thermal grease is used between different contact areas. When sunlight is incident on the surface of the PCM composites, the modified graphene nanoparticles will absorb and convert solar irradiation into thermal energy, and the converted thermal energy is applied to power the TE generator to generate electricity. The cooler temperature is maintained at 10 °C to dissipate the transferred thermal energy. In the control experiment, we use a pure PCM thermal pack to replace the PCM composites.

Figure 5b presents the variation of generated output voltage with time at a solar illumination of 5 kW/m^2^. As shown, the solar–electric conversion system with the PCM composites can produce a maximum output voltage of 0.14 V. For the control experiment system, the maximum output voltage is less than 0.04 V. In addition to the generation of output voltage, as the solar simulator is turned off, both the generated output voltage of the two systems will quickly drop. The system with the PCM composites is able to continuously produce electric energy for 620 s with the solar simulator being turned off. For the control system, the lasting duration is only 320 s, which is much less than that in the system with the PCM composites. We also test the stability of the device, and the results are shown in Appendix A. It can be seen that the device can generate a maximum output voltage of 0.13 V after continuously running for 20 cycles. Therefore, compared with pure paraffin, the prepared PCM composites are of great interest in the field of solar energy utilization.

## 4. Conclusions

In conclusion, novel form-stable solar thermal storage materials were proposed by embedding paraffin PCMs within porous G–Cu heterostructures. The porous G–Cu heterostructures were developed via sintering multilayer copper meshes, and they provided a high heat-conducting network, which enabled the rapid transfer of converted thermal energy, thereby improving the heat transfer capability of composites. The thermal conductivity of our prepared PCM composite was up to 2.99 W/(m K). Furthermore, graphene nanoparticles that were modified on the surface were capable of directly absorbing and storing solar energy while efficiently preventing leakage during the phase change process. Based on the advantages, such as high solar energy absorption, high thermal conductivity, and anti-leakage properties, the prepared PCM composites were demonstrated to be suitable for solar–electric systems, which had higher solar–electric conversion efficiency and longer electricity supply time compared with the system with pure paraffin. Considering the general feasibility of the method to prepare various kinds of PCM composites through our direct and simple synthesis process, it is expected that the PCM composites will have an extensive application value to expand solar utilization and other solar-related fields.

## Figures and Tables

**Figure 1 polymers-15-04723-f001:**
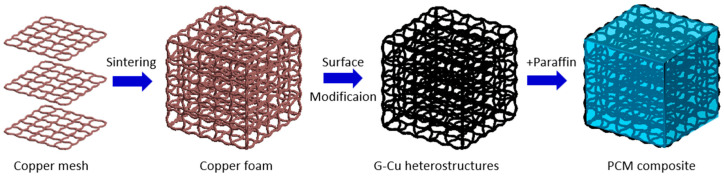
Synthesis process of the graphene–copper-based PCM composites.

**Figure 2 polymers-15-04723-f002:**
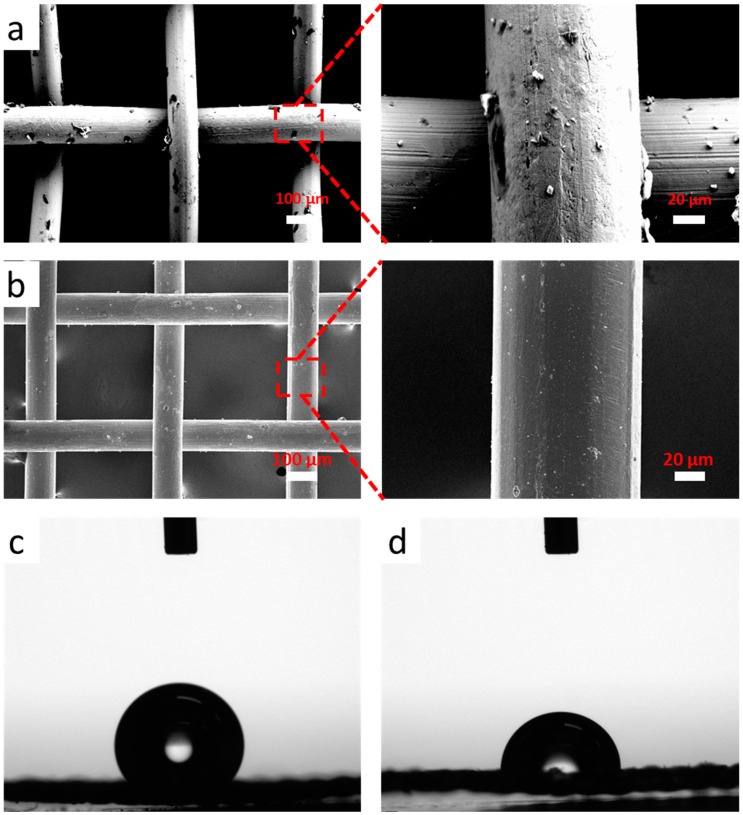
Characterization of the G–Cu heterostructures and untreated copper mesh. SEM images of the (**a**) G–Cu heterostructures and (**b**) untreated copper mesh at different magnifications. Photograph of contact angle measurement of (**c**) the modified G–Cu heterostructures and (**d**) untreated copper mesh.

**Figure 3 polymers-15-04723-f003:**
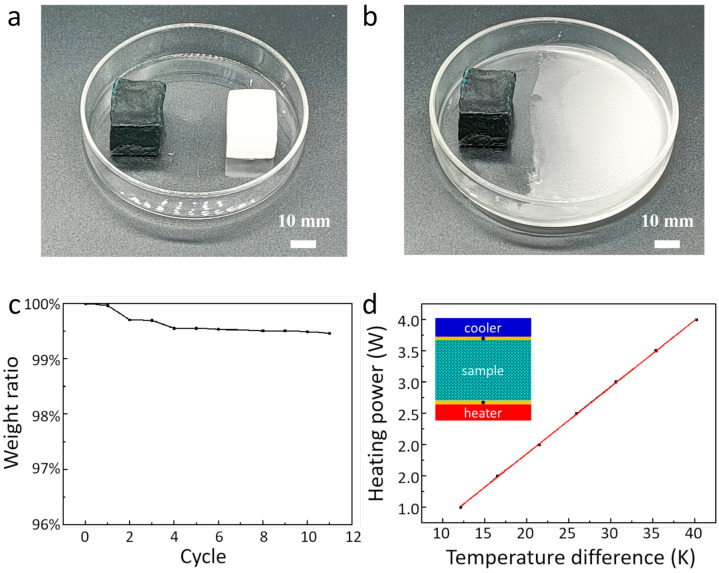
Measurement of the thermal stability and thermal conductivity of PCM composites. (**a**) Optical images of the PCM composites and pure paraffin block before heating. (**b**) Optical images of the PCM composites and pure paraffin block after heating for 10 min. (**c**) Leakage measurements of the PCM composites. (**d**) Thermal conductivity measurement of the PCM composites. The inset shows the schematic of a differential steady-state method.

**Figure 4 polymers-15-04723-f004:**
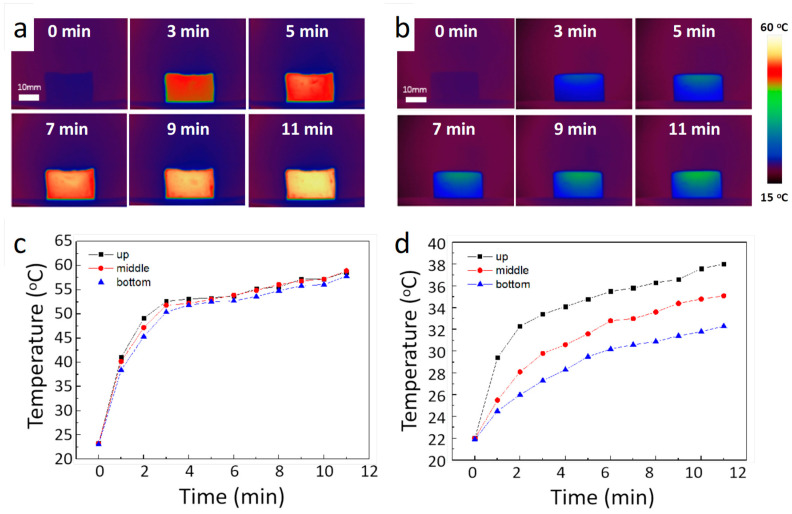
Solar charging process. Time-sequential IR images of (**a**) the PCM composites and (**b**) pure paraffin thermal pack with different solar charging times. Temperature variation of (**c**) the PCM composites and (**d**) pure paraffin thermal pack.

**Figure 5 polymers-15-04723-f005:**
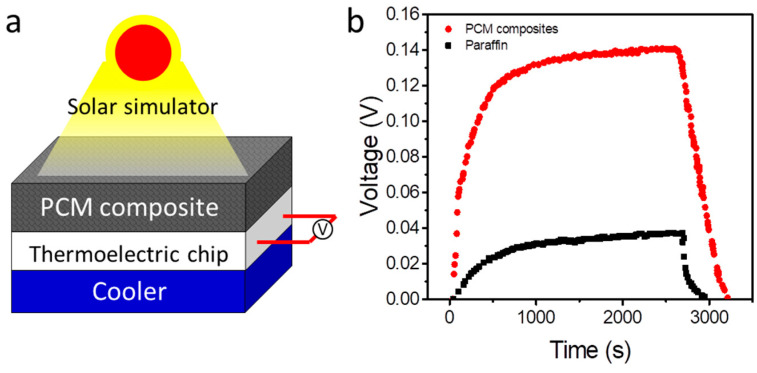
Solar–electric conversion performance. (**a**) Schematic of solar–electric conversion device based on PCM composites. (**b**) Output voltage of different systems under simulated solar irradiation.

## Data Availability

The data presented in this study are available on request from the corresponding author.

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
