# Peer review of "Form-Stable Composite Phase Change Materials Based on Porous Copper–Graphene Heterostructures for Solar Thermal Energy Conversion and Storage"

_polymers, 2023, doi:10.3390/polym15244723_

Round 1
Reviewer 1 Report
Comments and Suggestions for Authors
The manuscript entitled: "Form-Stable Composite Phase Change Materials based on Porous Copper-Graphene Heterostructures for SolarThermal Energy Conversion and Storage” present results from a great interest field. However, several points need to be addressed in this manuscript:
The paragraph from page 4 line 164-175 resumes the preparation method of G-Cu heterostructure. I suggest the author to include the text and figure 1 in paragraph 2.2.
I recommend the authors to specify the condition for determining wettability and, in figure 2 caption, the time at which the photo was taken.
Do the authors study if the average thickness of the graphene layer influences the properties of the solar-electric conversion devise and how many runs this devise can work in the mentioned parameters?
In order to have a better image of the novelty, in the Results and Discussions chapter, the authors must compare the results obtained with those from the literature, some of them being mentioned in the Introduction
Author Response
The manuscript entitled: "Form-Stable Composite Phase Change Materials based on Porous Copper-Graphene Heterostructures for Solar Thermal Energy Conversion and Storage” present results from a great interest field. However, several points need to be addressed in this manuscript:
- The paragraph from page 4 line 164-175 resumes the preparation method of G-Cu heterostructure. I suggest the author to include the text and figure 1 in paragraph 2.2.
Reply: Thanks for the comment. In the revised version, we rewrote the section, and added the text and Figure 1 in paragraph 2.2.
- I recommend the authors to specify the condition for determining wettability and, in figure 2 caption, the time at which the photo was taken.
Reply: Thanks for the comment. In the revised version, we added more description on the measurement condition for determining wettability, and described the time at which the photo was taken. The description was as follows:
“To reduce the contact angle hysteresis influence caused by the water droplets, we placed the needle to a distance of 1 to 2 mm up to the test samples. A computer was used to control the machine to drop 2 μL of water on the surface of the samples. A calculator continuously shot with a high-power camera, meanwhile the computer automatically calculated the contact angle. During the test, the room temperature was 20 â—¦C and relative humidity was 30 %.”
- Do the authors study if the average thickness of the graphene layer influences the properties of the solar-electric conversion devise and how many runs this devise can work in the mentioned parameters?
Reply: Thanks for the comment. The graphene layers were able to directly absorb solar radiation and convert it into thermal energy. In previous reports, many researchers investigated the effect of the CH4 flow rate and reaction time affected on the thickness of graphene grown on the copper. Here, we prepared copper-graphene heterostructures under a suitable reaction condition, which can form a uniform graphene layer on the surface of copper.
The prepared PCM composite presented excellent thermal performance and form stability. Figure 3a and Figure 3b presented the measurement results of the form stability of the PCM composites. As shown, after being heated for 10 min, our prepared PCM composites remained its original shape, and there were no melted liquid paraffin to be observed to flow out of the sample. In contrast, under the same heat condition, the commercial paraffin PCM block was completely melted into liquid within 60 s. We also evaluated the form stability of the PCM composite by measuring the weight change of the sample after repeated heating and cooling. No obvious weight change of the sample was observed after continuously running for 10 cycles. In the revised version, we also tested the stability of the device, and the results was shown in Figure S3. It can be seen that the device could generate a maximum output voltage of 0.13 V after continuously running for 20 cycles
- In order to have a better image of the novelty, in the Results and Discussions chapter, the authors must compare the results obtained with those from the literature, some of them being mentioned in the Introduction
Reply: Thanks for the comment. In the revised version, we added Table 1 which summarized the results obtained with those from the literature. Thanks again for the kind suggestion and great help.
Reviewer 2 Report
Comments and Suggestions for Authors
The manuscript entitled “Form-Stable Composite Phase Change Materials Based on
Porous Copper-Graphene Heterostructures For Solar Thermal Energy Conversion and Storage” has been submitted by authors. Some issues to be addressed which will improve the quality of manuscript. Therefore, I recommend this work could be published after the major revision
1) In Introduction part, need to add new paragraph with comparative result of recent study.
2) The experimental section preparation of graphene nanoparticles is not clear and needs to be rewritten.
3) There are some mistakes, so the language of the manuscript needs to be revised.
4) The structural characterization of the preparation of Porous Copper-Graphene Heterostructures is notably absent. The inclusion of X-ray diffraction (XRD), Fourier-transform infrared spectroscopy (FTIR), and Raman spectroscopy techniques would be beneficial for the author to confirm the formation of heterostructures.
5) The author should consider enlarging the font size of Figures 3 and 4 (c, d) to enhance readability for the readers.
6) The author is required to provide an in-depth explanation of the solar simulator light source, including its light intensity.
Comments on the Quality of English Language
Moderate editing of English language required
Author Response
The manuscript entitled “Form-Stable Composite Phase Change Materials Based on Porous Copper-Graphene Heterostructures For Solar Thermal Energy Conversion and Storage” has been submitted by authors. Some issues to be addressed which will improve the quality of manuscript. Therefore, I recommend this work could be published after the major revision
- In Introduction part, need to add new paragraph with comparative result of recent study.
Reply: Thanks for the comment. In the revised version, we added Table 1 with comparative result of recent study, and also cited more recent references.
- The experimental section preparation of graphene nanoparticles is not clear and needs to be rewritten.
Reply: Thanks for the comment. In the revised version, we rewrote this section preparation of graphene nanoparticles.
- There are some mistakes, so the language of the manuscript needs to be revised.
Reply: Sorry for the edits and minor questions. In the revised version, we double-check the whole manuscript and corrected all the edits and minor questions. Thanks again for the kind suggestion and great help.
- The structural characterization of the preparation of Porous Copper-Graphene Heterostructures is notably absent. The inclusion of X-ray diffraction (XRD), Fourier-transform infrared spectroscopy (FTIR), and Raman spectroscopy techniques would be beneficial for the author to confirm the formation of heterostructures.
Reply: Thanks for the comment. In the revised version, we analyzed the structural characterization of the preparation of porous copper-graphene heterostructures by X-ray diffraction (XRD), and Raman spectroscopy techniques, as shown in Figure S1 and Figure S2. We also added the following description.
“To further analyze its structure, the X-ray diffraction (XRD) and the Raman spectrum were acquired from the outer surface under 532-nm laser excitation as shown in Figure S1 and S2. From the XRD results in Figure S1, it can be seen that there were peaks at 2θ degree of about 43o, 50o and 74o which were well indexed to (111), (200), and (220) for Cu, respectively. The peak at 2θ degree of 26.3o was indexed to the C (002), indicating that the composites only consisted of graphene and copper, where graphene was composed of crystalline carbon atoms. From its Raman spectrum, we could see the typical graphene Raman peaks (G-band at ~1342 cm-2, D-band at ~1580 cm-2, and two-dimensional band at ~2680 cm-2) appearing on the surface.”
- The author should consider enlarging the font size of Figures 3 and 4 (c, d) to enhance readability for the readers.
Reply: Thanks for the comment. In the revised version, we modified Figures 3 and 4 (c, d) with a large font size.
- The author is required to provide an in-depth explanation of the solar simulator light source, including its light intensity.
Reply: Thanks for the comment. The solar irradiation was produced by a solar simulator (CEL-PE300L-3A, China), and a solar power meter (CEL-NP2000-2A, China) was used to calibrate the solar power density. During the experiment, the solar power intensity was adjusted to 5 kW/m2 to achieve rapid solar charging. In the revised version, we provided more description on the solar simulator light source.
Round 2
Reviewer 2 Report
Comments and Suggestions for Authors
The author addresses all comments very carefully, and I highly recommend accepting the work in its present form.